# Nutritional analysis of commercially available, complete plant- and meat-based dry dog foods in the UK

**Rebecca A. Brociek[1]\***, **Dongfang Li[2]**, **Richard Broughton[3]**, **David S. Gardner**[1]\*

**1** School of Veterinary Medicine and Science, University of Nottingham, Nottingham, United Kingdom,
**2** School of Biosciences, University of Nottingham, Nottingham, United Kingdom, **3** Institute of Aquaculture, University of Stirling, Stirling, Scotland

\* rebecca.brociek@nottingham.ac.uk (RAB); david.gardner@nottingham.ac.uk (DSG)

## Abstract

### Background

Adoption of a plant-based diet is a popular lifestyle choice for many owners of canine companion animals. Increasingly, owners would like to feed their canine companions a similar diet. A plant-based dietary pattern has been reported to be associated with some micronutrient deficiencies. Complete dog foods are, by definition, supposed to be nutritionally replete in all macro- and micronutrients. Few studies have reported a full nutritional analysis of complete, dry plant- versus meat-based dog foods.

### Method

31 dry commercially available dog foods (n = 19 meat-based, n = 6 veterinary and n = 6 plant-based) were analysed for total protein content and individual amino acids, fatty acids, major and trace elements, vitamin D and all B-vitamins.

### Results

Nutritional composition of meat and plant-based foods were nutritionally similar, except for iodine and B-vitamins, which were lower in plant-based foods. The majority (66%) of veterinary diets with lower total protein by design, were also deficient in one or more essential amino acids. Isolated instances of non-compliance to nutritional guidelines were observed across all food-groups. Of the tested nutrients 55%, 16%, 24% and 100% of foods met all amino acid, mineral, B-vitamin, and vitamin D guidelines, respectively.

### Conclusions

Adopting a plant-based dietary pattern for your companion canine can provide nutritional adequacy with respect to the majority of macro- and micronutrients, with the

**Data availability statement:** All anonymized data for the products used in this manuscript are available from the corresponding authors upon reasonable request or via The University of Nottingham research data repository at http://doi.org/10.17639/nott.7586. Any individual company requesting data on any of their products used in this manuscript, will be provided to them on an individual non-anonymised basis.

**Funding:** This study was funded by the Biotechnology and Biological Sciences Research Council (BBSRC) as part of the University of Nottingham Doctoral Training Partnership (DTP) PhD studentship awarded to R.A.B with D.S.G. as Principal Supervisor (Grant code: RS86P5). The funders had no influence on study design, data collection and analysis, decision to publish, or preparation of the manuscript.

**Competing interests:** DSG conducts nutritional analysis for one of the tested companies' foods. D.S.G is not involved in the formulation of the food, nor has influence on the design or reporting of results. Since completing the analysis, R.A.B has purchased shares in the same company; testing was completed before this occurred. This does not alter our adherence to PLOS ONE policies on sharing data and materials. There are no patents, products in development or marketed products associated with this research to declare.

exception of iodine and B-vitamins, which could easily be supplemented. Veterinary-renal diets, purposely low in crude protein, often have less than optimal essential amino acid composition. These data provide important new information for owners of companion canines being fed plant-based or veterinary diets.

## Introduction

Veganism is increasingly becoming a popular dietary choice for many people, whether it be for health reasons or concerns for animal welfare and/or the environment. The number of self-declared vegans in the UK quadrupled between 2014 and 2019, from 150,000–600,000 (0.25% to 1.2% of the population [1]. Many share their homes with their omnivorous, canine companions (Bosch et al., 2015). Owners of companion animals, who identify as vegetarian or vegan therefore face an ethical dilemma – should they feed animals to their animals? [2]. Consequently, there has been an increase in the availability of 'complete' plant-based pet foods on supermarket shelves with little to no independent assessment of their nutritional soundness.

Meat-based food, including the incorporation of by-products from the meat industry, has long been seen as the 'natural' way to feed companion canines. Meat is high in protein and thus, provides the building blocks of protein via 'proteogenic amino acids (AA)', which are classified as either non-essential, conditionally-essential, or essential (EAA) [3]. The distinction being whether or not the body can form the amino acid from other substrates, for example by transamination (i.e., 'non-essential'), whether the amino acid only becomes essential during certain high-demand 'conditions' such as pregnancy ('conditionally-essential') or whether the amino acids cannot be made in the body and thus must be acquired and ingested in the diet ('essential'). In addition to protein, meat, dairy and other animal by-products tend to also be high in B-vitamins, selenium and organic phosphorous [4]. For such a diet to be labelled as 'complete' and to adhere to nutritional guidelines, in the UK/EU, administered by the Fédération Européenne de l'Industrie des Aliments Pour Animaux Familiers (FEDIAF – the European Pet Food Industry Federation) or, in the USA, by the Association of American Feed Control Officials (AAFCO), supplemental micronutrients are always added, as listed on the label. Dairy protein – casein – for example, is deficient in methionine, as are many other individual sources of plant-based protein; soybeans, beans and chickpeas are similarly unbalanced, lacking single amino acids from their profile, but when used in combination, can be effective for delivering sufficient amino acids and crude protein. Nevertheless, the few studies to date that have assessed the nutritional completeness of plant-based pet foods (sold in either Brazil [5] or Canada [6]) reported multiple nutritional deficiencies in their composition.

In the UK, a survey of dog owners reported that the most important attributes any alternative diet would need to provide were 'confidence about nutritional soundness' and 'confidence about pet health' (cited by 84% and 83% of these respondents, respectively [7]). Similar observations were made in a separate study in North America; of those owners that did not already feed plant-based diets, a large proportion

(45%; 269/599) were concerned, or wanted more information, about the nutritional completeness/adequacy of plant-based pet food [6]. Indeed, 74% stated that this was the primary reason that they didn't currently feed a plant-based pet food. Other studies have drawn varying conclusions regarding the nutritional appropriateness of feeding companion canines a plant-based diet. One study concluded that further research was required to determine if long-term feeding of plant-based diets can meet and maintain amino acid and other nutrient targets in canines [8]. Others found isolated instances of various micronutrient deficiencies in vegan foods available in Brazil [5]. Further independent assessment is therefore warranted to provide evidence for owners of companion canines, that feeding a plant-based diet can provide nutritional completeness.

The primary objective of the current study was to measure the nutritional composition of complete, dry, meat- and plant-based foods available for canines on the UK market. We hypothesised that plant-based foods would be: 1) less 'nutritionally-complete' with lower compliance to nutritional guidelines than meat-based foods, 2) have lower protein and branched-chain amino acid content and 3) have lower B-vitamin, particularly B12, content. Foods indicated by a veterinarian to be fed to companion animals with kidney (e.g., chronic kidney disease, CKD) or urogenital problems (e.g., tendency to kidney stones) are, by design, low in protein but have a guaranteed analysis and are thus also labelled as 'complete'. We hypothesised that 4) such 'veterinary-renal' foods would be low in essential amino acids relative to comparable 'non-renal' meat-based foods.

## Materials and methods

*Selection of dog food:* Thirty-one complete, dry kibble dog foods were acquired over a 4-week period (September 2022) from either online or high-street pet supermarkets in the UK, representing twenty-seven different brands. Exclusion criteria included not being a 'complete' food, not being labelled as for either 'adult' or 'adult and senior' dogs, foods that weren't readily available to the public (i.e., required a prescription) and foods that did not come in packages of 3 kg or less, to limit food waste. Foods were grouped according to their main protein source as listed on the label: "meat-based" (n = 19; including, poultry (n = 7), lamb (n = 6) and beef (n = 6); "plant-based", including vegan (n = 4) and vegetarian foods (n = 2). These six plant-based foods were the only brands available meeting our criteria sold in the UK, at the time of purchase. In addition, six guaranteed-analysis veterinary diets (n = 6) were selected that were marketed as 'reduced protein', specifically for dogs with renal and/or urinary tract problems ('Veterinary-renal' foods). No specific flavour was declared for these foods.

*Preparation of dog food:* Bags were inverted several times before opening. Approximately 100g of each dry food was sampled and frozen for a minimum of 24h at −20°C, before freeze-drying for 48-72h (Scanvac Coolsafe, Labogene, Denmark). Dry samples were then milled using a centrifugal mill (ZM 200 Ultra Centrifugal Mill & Vibratory Feeder DR100, Retzsch, Germany) at 10,000 RPM to a homogenous powder and stored in a 250 ml polypropylene container at −20°C until required for analysis. Foods were not randomised or blinded for the authors. During preparation, all foods were given a sample number between 259 and 300 which was provided to analysts and technicians handling the samples, blinding these individuals to the provenance of each sample. All 31 complete dry dog foods were tested singularly and compared to European Pet Food Industry Federation (FEDIAF) guidelines for maintenance of adult dogs per 1000kcal of metabolisable energy (ME), at a maintenance energy requirement (MER) of 110kcal/kg$^{0.75}$ body weight, necessary for dogs of moderate activity level (1–3hrs/day) [9].

*Protein and moisture content:* Moisture content was determined by weighing a known quantity (100-200g) of frozen food, freeze-drying to completeness under vacuum for at least 2 days and re-weighing. Crude protein was determined using FlashEA® 1112 N/Protein (Thermo Fisher Scientific™) Nitrogen and Protein Analyzer, by directly measuring nitrogen content [10] and subsequently multiplying by 6.25. This method makes multiple assumptions about the nitrogen in the diet and can result in an over-estimation for vegetable proteins. This is often not a problem for mixed diets (FAO, 2003), such as dog food, however, the sum of individual amino acids was also measured, and the values compared.

***Amino acid analysis, preparation of sample:*** Samples containing approximately 5 mg nitrogen were oxidised in 20 ml headspace glass crimp neck vials with 2.5 ml of chilled, freshly made oxidation solution (10% of hydrogen peroxide (30% v/v) incubated (1 hour at 20–30 °C) in 87% v/v formic acid with 0.55% (w/v) phenol as an oxygen scavenger) for 16–18 hours at 4°C. After oxidation, 0.42g of sodium metabisulphite was added to decompose any excess oxidation reagent. The samples were then hydrolysed with 2.5 ml 12M HCl and 0.5 ml of 6M HCL containing 1% phenol (w/v). The contents were mixed on a vortex mixer until all the sample was finely distributed in the acid. After mixing, the solutions were hydrolysed at 110°C for 24 h in an air draft oven. Sample pH was adjusted with 4M ammonium formate to 2.75 and made up to a volume of 50 ml with 20mM pH2.75 ammonium formate buffer. After centrifugation at ×3000g for 2 minutes, 1–2 ml supernatant was passed through 0.22μm filter. The filtrate was diluted accordingly with ammonium formate buffer and an internal standard mix compromising cell-free stable isotope labelled ($^{13}$C,$^{15}$N) target amino acids to adjust the final amino acid values in the sample to be within the calibration range (nitrogen, 1–10 μg/mL) of the instrument. A standard reference material of soy flour (National Institute of Standards and Technology, Maryland, USA, SRM 3234) was analysed in parallel to validate the accuracy of the hydrolysis and the amino acid analysis.

***Amino acid analysis; uHPLC-MS/MS:*** An aliquot of 200 μL was dispensed into HPLC vials followed by 200 μL of an internal standard comprising cell-free isotopically labelled ($^{13}$C, $^{15}$C) target amino acids (S1 Table in S1 File). An amino acid standard curve was generated using Supelco amino acid standard mix, to which L-cysteic acid and methionine sulfone were added at a similar concentration with the amino acid standard mix. All samples were separated and analysed using a Thermo-Fisher Vanquish (uHPLC) and Altis Triple Quadrupole Mass spectrometer (MS/MS) with heated electrospray ionization (H-ESI) system. Positive ion mode was used for all amino acids. 1 μl was injected on a Thermo Scientific™ Acclaim™ Trinity P1 mixed mode column (150 mm x 2.1 mm, 3μM) at 30 °C. Mobile phases consisted of ammonium formate in water at pH 2.75 for phase A and a mixture of ammonium formate (100mM) in water and acetonitrile (80/20 v/v) for phase B. Chromatographic separation was achieved by gradient elution with MRM transition conditions as described (S2 Table in S1 File). Sheath gas was set at 45 arbitrary units, auxiliary gas at 15 arbitrary units, and spray voltage at 3500 V for positive ionization. Vaporizer temperature was set to 370 °C and transfer tube temperature to 270 °C, while source fragmentation was applied at 15 V. Data was acquired in Multiple Reaction Monitoring (MRM) mode using a resolution of 0.7 full width at half maximum (FWHM) for both quadrupoles. All compounds were detected in positive-ion mode.

***Amino Acid reporting:*** Nine out of ten essential amino acids were analysed in the current study, as tryptophan could not be detected using our current methods. Serine data are missing for nine samples, thus remaining data are reported for information only (S3 Table in S1 File). Additionally, the oxidation step of the preparation method leads to conversion of methionine to methionine sulfone and cysteine to cysteic acid ([11], therefore results for methionine sulfone and cysteic acid will be reported as methionine and cysteine, respectively. TraceFinder Version 4.1 was used to analyse raw data. Curve shape was standardised to ensure comparability between sample analyses. Conversion from reported units to g/1000kcal ME was completed using the calculated Atwater value for each individual food and the FEDIAF conversion value [9].

***Fatty acid methyl ester (FAME) lipid extraction and separation:*** 5mL of 0.6M sucrose extraction buffer (for composition, see S4 Table in S1 File) was added to ~2g dried food and homogenised using a GentleMACS tissue dissociator (Miltenyi Biotec, Ltd). 7mL of 0.6M sucrose extraction buffer was added to the homogenate, and the sample further centrifuged (Thermo Heraeus Pico 17, Thermo-Fisher Scientific™) at 3222 rpm (2000*g*) for 5 mins. The homogenate was layered for sucrose cushion extraction (1mL 0.6M sucrose extraction buffer, 1.5mL homogenate, 2.5mL 0.25M sucrose extraction buffer) before ultracentrifugation at 100,000*g* for 1 hour (Hitachi CP80NX ultracentrifuge, P55ST2 rotor). The upper lipid layer was removed and further extracted by adding 5mL 2:1 chloroform:methanol, vortexed for 15 secs before the addition of 1mL 1% NaCl and further vortexed until homogenous. The sample was then centrifuged at 1000*g* for 2 minutes, and the lower (chloroform with lipid) fraction was transferred to a new glass centrifuge tube. Any remaining lipid

was extracted by adding 3mL chloroform, vortexed for 20 seconds, centrifuged at 1000$g$ for 2 minutes and the bottom fraction pooled to the new glass centrifuge tube. The pooled sample was dried under nitrogen and resuspended in 400 $\mu L$ of hexane. To this suspension, 0.7mL 10M KOH and 5.3mL methanol were added, and the sample heated at 55°C for 90 min. After cooling, 0.58mL 12M $H_2SO_4$ was added and further incubated at 55°C for 90 min. After cooling, 3mL hexane was added, mixed and centrifuged. The upper hexane layer was removed and concentrated by drying under nitrogen and reconstituting in 400 $\mu L$ hexane. The sample was stored at −30°C until required for analysis by GC-MS analysis.

*FAME Gas Chromatography Mass Spectrometry (GC-MS):* The fatty acid methyl esters (1µl) were injected (split ratio 50:1) into a gas chromatograph (GC) (Trace 1300, Thermo Fisher Scientific™) coupled with mass spectrometer (MS) (ISQ 7000, Thermo Fisher Scientific™). Separation of fatty acid methyl esters was performed with a Varian CP-Sil 88 (100m length, 0.25 mm diameter, 0.20um film thickness, Agilent) capillary column with helium as carrier gas. Oven temperature (ramp up at 4°C/minute, from 140°C (hold for 5 minutes) to 240°C (hold for 10 minutes) and MS injector and transfer line temperature (260°C and 250°C, respectively) were preprogrammed. The ion source temperature set to 200°C. Characterization and identification of FAMEs was performed in scan mode. Quantification was completed by selective ion monitoring (SIM) mode of the most intense fragments. Data acquisition and processing were performed with the software Chromeleon (version 7.0, Thermo Fisher Scientific™).

*Mineral and trace elemental content analysis (ICP-MS):* Minerals and trace elements were determined by inductively coupled plasma-mass spectrometry (ICP-MS), expressed per unit dry weight (or Megacal), as previously described [12,13]. Briefly, approximately 0.2 – 0.3g of sample and 0.1 – 0.2g bovine liver (as certified reference material, CRM: 1577C [National Institute of Standards and Technology (NIST)]) were digested using 3mL nitric acid, 3mL deionised water (DI) and 2mL $H_2O_2$ in a digestion microwave (Multiwavepro, Anton Parr, settings: 12 tubes, 1000W, 45 mins). Digesta was transferred into 50mL centrifuge tubes, and an additional 7mL deionised water, used to rinse any remaining sample. 500 $\mu L$ of digesta was pipetted into polypropylene ICP tubes before ICP-MS analysis using an iCAP-Q (Thermo Fisher Scientific™) by the Department of Environmental Science, Faculty of Science, Sutton Bonington Campus, University of Nottingham. Using this method, 32 major and trace elements are reliably reported, with n = 13 referenced against FEDIAF guidelines [9]. Standardisation between batches was achieved by adjustment to the CRM with n = 23 elements reported.

*Vitamin analysis; Vitamin D:* 2-3g of freeze-dried food samples (n = 29) were sent to The Institute of Aquaculture, University of Stirling, UK. Vitamin D was analysed by LC-MS/MS using a Waters Xevo TQ-S mass spectrometer coupled to a Waters Acquity I class UPLC with an Acquity UPLC BEH C18 column. Briefly, 600 mg of dry, homogenised pet food was weighed into glass vials, with 30 µl of a 2.5 µg/ml deuterated cholecalciferol (D3) and ergocalciferol (D2) standard added to each sample, plus calibration standards. A calibration curve was processed at the same time as the samples (0–50 µg/ml of non-labelled D2, D3). Briefly, vitamin D in the foods was extracted using 4 ml of 1.5 M potassium hydroxide in ethanol, with pyrogallol as the antioxidant for 1 hour at 80°C, followed by extraction with 3 ml of hexane, with the addition of 3 ml of 1% (w/v) potassium chloride solution. Hexane extracts were transferred to clean glass vials, dried under nitrogen, then re-constituted in ethyl-acetate. Samples were then derivatized using 4-phenyl-1,2,4-tirazoline-3,5-dione (PTAD) for 1 hour prior to analysis by LC-MS (see S5 Table in S1 File).

*Vitamin analysis; Vitamins B1 - 12:* Approximately 2g of homogenised pet food (n = 17) were sent to Creative Proteomics Ltd, USA, and analysed for the full range of B-vitamins using an AB Sciex QTRAP® 6500 LC-MS/MS platform. Briefly, each sample was further ground on a MM 400 mill mixer for 5 min at a shaking frequency of 30 Hz. 100 mg of the homogenised powder was weighed into a 5 mL tube and homogenized at 30 Hz for 5 min, followed by 5 min ultrasonication in a water bath, then centrifuged at ×15,000$g$ for 10 min. An internal standard (IS) of riboflavin (B2)-13C2/15N, nicotinamide (B3)-d4 and nicotinic acid (B3)-d4 was prepared in 65% acetonitrile. Serially diluted calibration solutions containing the 10 targeted vitamins were prepared in the IS solution. 10 µL aliquots of the clear supernatants and the standard solutions were injected into a HILIC column (2.1*100 mm, 1.7µm) to run UPLC-MRM/MS with negative-ion mode on an Agilent 1290 UHPLC system coupled to an Agilent 6495C MS instrument, for detection and quantitation of ascorbic

acid, nicotinic acid, vitamin B5 and B7, or with positive-ion mode for detection and quantitation of B9 and B12. For quantification of vitamin B1, B2 and nicotinamide, the sample solutions were diluted 10-fold with the IS solution before injection. The mobile phase was 2 mM ammonium acetate (A) and acetonitrile (B) for gradient elution (90% to 10% B in 12 min), at 0.3 mL/min and 40°C. Concentrations of the detected vitamins were calculated by interpolating the constructed linear regression curves of individual compounds, with the data acquired from injections of the sample solutions, in an appropriate concentration range for each metabolite (example trace for Vitamin B12, S1 Fig in S1 File). Limits of quantification for each of the B-vitamins are reported in S6 Table in S1 File.

*Statistical analysis:* Data were analysed using analysis of variance (ANOVA) for the fixed effect of the three diet-types (meat-based or plant-based, veterinary). In order to meet assumptions for analysis by ANOVA, all data were checked for a normal distribution of residuals and respective Q-Q plots. If necessary, non-normally distributed data were log-transformed ($\log_{10}$) prior to analysis by ANOVA, or an alternative suitable non-parametric, distribution-independent test was used (e.g., Kruskall-Wallis NP-ANOVA). All such data were analysed using GraphPad Prism v9.5.0 (GraphPad Software Inc., California, USA) and GenStat v22 (VSNi Ltd., Rothamsted, UK). Since multiple amino acids, fatty acids and trace element data were derived from each sample, many may often show concordance between analytes in the same sample. To mitigate such over-dispersion, multivariate, linear discriminant analysis was used as an objective means to effectively demonstrate significant patterns in complex (i.e., multiple variates), potentially non-independent data using orthogonal partial least squares-discriminant analysis (OPLS-DA; SIMCA-P v19, Umetrics, Umea, Sweden). Certain analyses and graphical representations were also conducted in the open-source software JASP Team (v0.17.1.2023; jasp-stats.org), as indicated in appropriate Figure or Table legends.

*Data availability:* All anonymized data for the products used in this manuscript are available from the corresponding authors upon reasonable request or via The University of Nottingham research data repository at http://doi.org/10.17639/nott.7586. Any individual company requesting data on any of their products used in this manuscript, will be provided to them on an individual non-anonymised basis.

## Results

***Protein and amino acid content of foods***: Direct analysis of crude protein content of all foods indicated similarity in protein content between meat- and plant-based foods, with veterinary foods being, by design, lower in total protein (Fig 1a). The directly measured, versus stated protein content on the label, corresponded well (Fig 1b,1e). Measurement of all individual amino acids, including 9/10 essential for canines (Arginine, Histidine, Isoleucine, Leucine, Lysine, Methionine, Phenylalanine, Threonine and Valine) are also reported and data, as expected, were similar to total protein (Fig 1c). Compared across food types, then veterinary-renal diets generally had significantly reduced content of all amino acids (Table 1, for values per 100g DM). Nevertheless, essential amino acids are essential and must meet minimum inclusion levels; it was therefore notable that while 17/31 (55%, n = 11/19 meat-based, n = 2/6 veterinary, n = 4/6 plant-based) foods met EAA minimum inclusion levels, many veterinary foods did not (Fig 1e) – four of six being below nutritional guidelines for essential amino acids, with threonine below guideline inclusion levels in all four (range: 1.04-1.14g/1000kcal, FEDIAF: 1.3g/1000kcal; Fig 1e). Remarkably, one of the six veterinary-renal foods was below guideline amounts in 6/8 essential amino acids (isoleucine, leucine, methionine sulfone, phenylalanine, threonine and valine; Fig 1e). In an unbiased, multivariate analysis of all foods and all measurable amino acids, the three food types were clearly distinguishable from each other (Fig 1d). Most of the variation (68%) was explained by less total protein and individual amino acid content in veterinary foods (Fig 1e).

***Fatty acid content of foods:*** Broadly, the fatty acid composition of meat-based and veterinary foods was similar. Meat-based foods had the highest proportion of animal-based saturated fatty acids such as palmitic, stearic and arachidic acid (sum of saturated fats; meat-based, 67.2 ± 19.7; plant-based, 42.2 ± 5.1; veterinary, 56.6 ± 11.5 gms %; $P = 0.01$). Plant-based foods had the highest incorporation of mono- and poly-unsaturated fatty acids; oleic, linoleic and linolenic acid

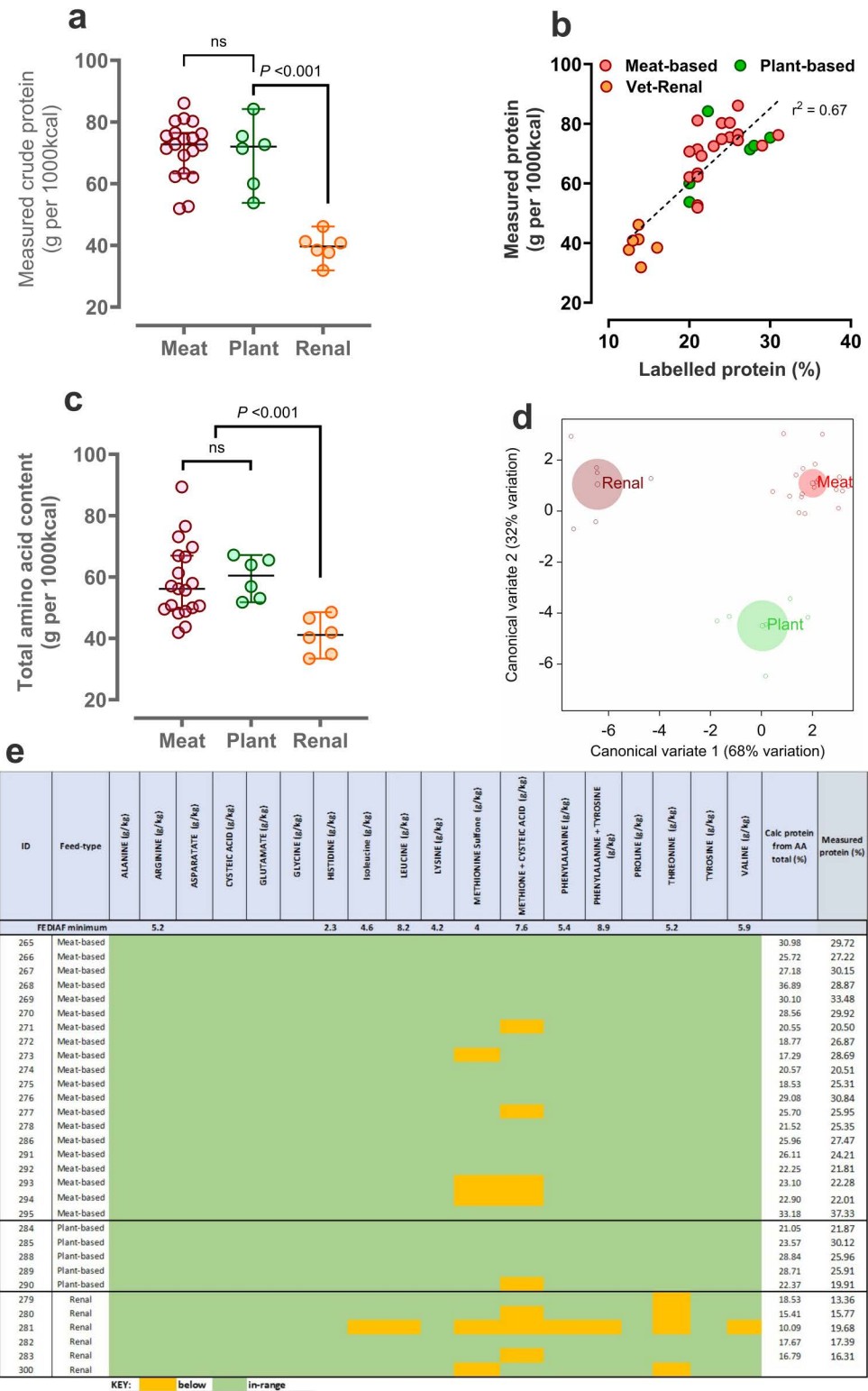

**Fig 1. Protein and amino acid content of dry feeds for dogs according to feed-type.** a) Individual data points for directly measured total protein, b) relationship between measured and labelled total protein, c) directly measured total alpha amino acids according to the three feed types, d) multivariate

analysis (discriminant plot) with all feeds (n = 31) and all measured amino acids (n = 19) represented, e) data expressed relative to the guideline nutritional minimum for each individual amino acid ('FEDIAF minimum'. **Green** boxes represent values in range according to our analyses, **yellow** boxes are below nutritional minimum.

**Table 1. Amino acid content of dry feeds for dogs according to feed-type, per unit dry matter.**

| Amino acid | FEDIAF (per 100g DM) | Meat-based (n = 19) | Plant-based (n = 6) | Renal (n = 6) | Statistics *P-value |
|---|---|---|---|---|---|
| Alanine (g) | – | 1.47 ± 0.06[a] | 1.31 ± 0.11[a] | 0.77 ± 0.11[b] | **<0.001** |
| Arginine (g) | 0.52 | 1.50 ± 0.06[a] | 1.46 ± 0.11[a] | 0.73 ± 0.11[b] | **<0.001** |
| Aspartate (g) | – | 2.28 ± 0.12[a] | 2.51 ± 0.22[a] | 1.44 ± 0.22[b] | **0.02** |
| Cysteic acid (g) | – | 2.93 ± 0.75[a] | 3.62 ± 0.42[b] | 2.53 ± 0.36[a] | **0.004** |
| Glutamate (g) | – | 3.58 ± 0.15[a] | 4.41 ± 0.27[a] | 2.16 ± 0.27[b] | **<0.001** |
| Glycine (g) | – | 2.22 ± 0.03[a] | 1.44 ± 0.18[a] | 0.95 ± 0.18[b] | **<0.001** |
| Histidine (g) | 0.23 | 0.49 ± 0.02[a] | 0.58 ± 0.03[a] | 0.32 ± 0.03[b] | **<0.001** |
| Isoleucine (g) | 0.46 | 0.89 ± 0.04[a] | 0.99 ± 0.07[a] | 0.59 ± 0.07[b] | **<0.001** |
| Leucine (g) | 0.82 | 1.85 ± 0.11[a] | 2.10 ± 0.20[a] | 1.32 ± 0.20[b] | **0.03** |
| Lysine (g) | 0.42 | 1.28 ± 0.07 | 1.32 ± 0.12 | 0.99 ± 0.12 | 0.12 |
| Methionine sulfone (g)[1] | 0.40 | 0.49 ± 0.02 | 0.49 ± 0.04 | 0.45 ± 0.04 | 0.77 |
| Methionine + Cysteic acid (g)[1] | 0.76 | 0.78 ± 0.03 | 0.85 ± 0.06 | 0.71 ± 0.06 | 0.23 |
| Phenylalanine (g) | 0.54 | 1.01 ± 0.05[a] | 1.19 ± 0.09[a] | 0.62 ± 0.08[b] | **<0.001** |
| Phenylalanine + Tyrosine (g) | 0.89 | 1.64 ± 0.08[a] | 2.05 ± 0.14[a] | 1.03 ± 0.14[b] | **<0.001** |
| Proline (g) | – | 1.67 ± 0.07[a] | 1.42 ± 0.14[a] | 0.74 ± 0.14[b] | **<0.001** |
| Threonine (g) | 0.52 | 0.84 ± 0.04[a] | 0.87 ± 0.07[a] | 0.46 ± 0.07[b] | **<0.001** |
| Tyrosine (g) | – | 0.63 ± 0.03[a] | 0.85 ± 0.06[a] | 0.42 ± 0.06[b] | **<0.001** |
| Valine (g) | 0.59 | 1.20 ± 0.05[a] | 1.23 ± 0.08[a] | 0.69 ± 0.08[b] | **<0.001** |

Table 1. All data are mean ± SD, units per 100g DM, based on 110kcal/kg maintenance energy requirement (MER). FEDIAF guideline content for reference. Tryptophan was not detectable by our methods. *differing superscripts in the same row indicate statistical significance at P < 0.05, as analysed by (non-parametric) one-way ANOVA.

(23.9 ± 12.5, 27.4 ± 8.6 and 3.43 ± 3.44 gms %, respectively). The majority of individual fatty acids in the pet foods analysed in this study were either saturated; C8:0 (caprylic – 23% of total), C16:0 (palmitic – 22% of total), C18:0 (stearic – 13% of total), C20:0 (arachidic – 1% of total) or unsaturated; C18:1n9 (oleic – 16% of total), C18:2n6c (linoleic – 14% of total), linolenic (C18:3n3 - 2% of total; Fig 2a). Whilst measurable, the combined sum of C6:0 (caproic), C10:0 (capric), C11:0 (undecanoic), C12:0 (lauric), C13:0 (tridecanoic), C14:1 (myristoleic), C15:0 (pentadecanoic), C17:0 (heptadecanoic), C18:1n9t (elaidic), plus other long-chain fatty acid derivatives (C20:0, arachidic – C24:1, nervonic) comprised <5% of total fat in each sample ('other' in Fig 2a). All foods were replete in linoleic acid, according to the nutritional guidelines [9], on a gram per 100g total lipid (i.e.g., % lipid mass) or mass-basis (i.e., ≥ 1.32g/100g DM; Fig 2b, d). Plant-based foods had significantly greater (P < 0.01 by Kruskall-Wallis NP ANOVA) linoleic acid (27.3 ± 8.6 gms %) than meat-based (9.89 ± 5.17 gms %) and veterinary (15.2 ± 3.1 gms %) foods (Fig 2b). Unbiased multivariate discriminant analysis showed a clear separation of plant-based from both meat-based and veterinary – which were similar with respect to fatty-acid composition – along the first principle component (73% variation explained) distinguishing plant-based diets as having an overall greater incorporation of caprylic (+1.24 contribution to latent vector 1) and linolenic acid (+0.26; Fig 2c). Using macronutrient data on the labels of each food to calculate gross and metabolisable energy, according to Atwater criteria, indicated that

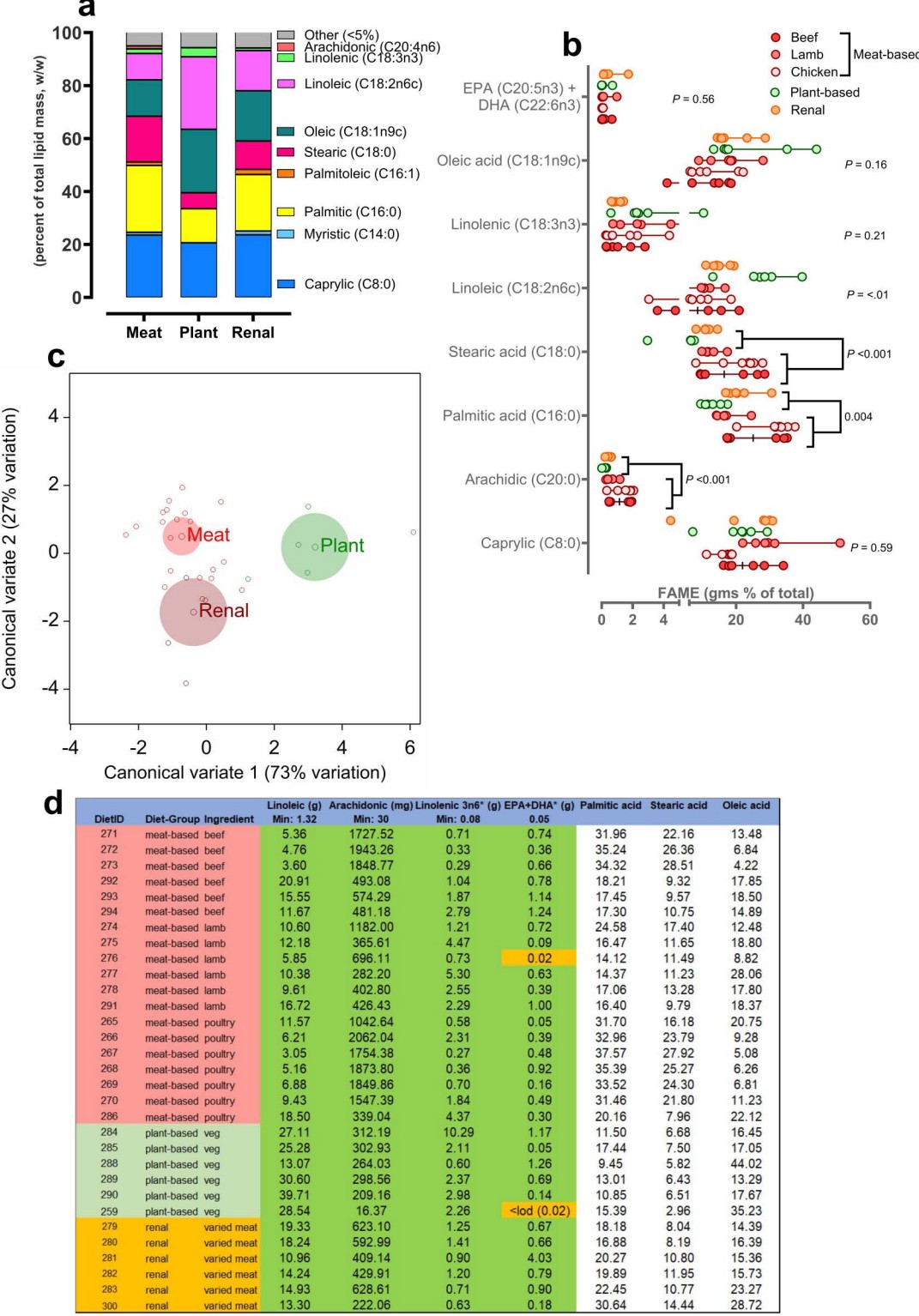

**Fig 2. Fatty acid content of dry feeds for dogs according to feed-type.** a) relative proportions (w/w %) of prevalent fatty acids between feed types, b) individual data points for measured fatty acids between feed types with beef, lamb and chicken combined as 'meat-based' for analysis by 1-way

ANOVA, c) multivariate analysis (discriminant plot) with all feeds (n = 31) and all measured fatty acids ≥LOD (n = 9) represented, d) data (individual measured value) expressed relative to the guideline nutritional minimum for each individual fatty acid ('Min'). **Green** boxes represent values in range according to our analyses, **yellow** boxes are below nutritional minimum. White boxes, no specified nutritional range.

veterinary foods had higher energy content than both meat or plant-based foods (Energy density: meat-based, 332 ± 16; plant-based, 328 ± 5; veterinary, 376 ± 10 kcal ME/100 g DM; $P < 0.001$), largely due to incorporation of more fat in the food (% fat on label: meat-based, 12.8 ± 3.0; plant-based, 10.4 ± 1.8; veterinary, 17.3 ± 1.7 gms fat as fed, $P < 0.001$).

*Major and trace minerals*: Whilst only 16% of individual foods tested (n = 5/31, all meat-based) satisfied all mineral guidelines, overall compliance was moderate-to-high (n = 342/391 [87%] of minerals were 'in-range', i.e., n = 13 guidelines × n = 31 foods = 391 in total, excluding Ca and P which are purposely reduced in renal foods; Fig 3a). The individual foods with 'out-of-range' minerals tended to be isolated instances (e.g., of chloride or zinc), but below recommended incorporation of iodine and selenium were common in all foods (Fig 3a, Table 2), whether expressed per unit mass (Fig 3b) or unit energy (Fig 3c, d). Veterinary diets formulated for dogs requiring renal support were lower in calcium ($P = 0.009$), phosphorous ($P = <0.001$), magnesium ($P < 0.001$), iron ($P = 0.006$) and selenium ($P = 0.01$; Table 2). Plant-based foods had greater potassium and lower iodine content than other food types (both P ≤ 0.01).

*Vitamins, Vitamin D*: All foods, when directly measured for vitamin D content, were within the recommended nutritional range (i.e., between FEDIAF nutritional minimum and maximum of 138–800 IU/1000kcal; Fig 4a). *B-vitamins*: 17 foods (meat-based, n = 8; plant-based, n = 5; veterinary, n = 4) were analysed for a full panel of B-vitamins (B1, B2, B3, B5, B6, B7, B9 and B12), of which seven (all except vitamin B7) have FEDIAF nutritional minimum recommendations. B-vitamin content of foods were mostly comparable between food types, but consistently lower B-vitamin content was noted in plant- versus meat-based foods for vitamins B3, B9 and B12 (Table 3; Fig 4b, c; all at $P ≤ 0.05$, 1-way NP ANOVA). Accordingly, when summated, plant-based foods had lower B-vitamin content than meat-based foods (Fig 4d, Table 3). Overall, compliance of foods to nutritional recommendations for B-vitamins was poor – only four of 17 tested (23%) met all minimum requirements for B-vitamins, with any deviation from recommended being below the guideline level (Fig 4e).

## Discussion

Adopting a plant-based dietary pattern is becoming increasingly common in Western society whether for health benefits [14,15] or for environmental considerations [16,17]. Canines, as omnivores, are well-adapted to receive such a diet. Vegetarians and vegans, by definition "plant-based", more commonly experience some micronutrient deficiencies [18,19], easily rectified through supplementation. For companion animals fed a 'nutritionally-complete' plant-based diet, such deficiencies should not occur. The Food Standards Agency, UK requires labelling of foods as 'complete' to mean that feeding such food would give the companion animal all the nutrients it requires for maintenance or for growth and development. Few studies have independently tested this assumption. Here, we show that 'complete', dry plant-based foods for canines were replete in protein and amino acids, but consistently low in iodine and some B-vitamins – similar to observations made for human populations following a vegetarian or vegan diet [18,19]. Interestingly, guaranteed-analysis, veterinary-renal diets designed for a particular nutritional purpose – thus, low in protein to support dogs with moderate kidney disease – were also low in many essential amino acids. No foods tested here met all nutritional guidelines, as previously described by us in regard to mineral content of a range of pet foods [13].

*Protein, amino acids and fatty acids*: Independent analysis of total crude protein correlated well with the sum of alpha amino acids measured in all foods, and the amount of protein as reported on food labels. The veterinary foods were, as expected, lower in measured total protein but, unexpectedly, were also relatively deficient (cf. guidelines for such foods) in a number of essential amino acids (EAA); 66% of the foods were low in at least one EAA despite us not reporting relatively low levels of methionine due to analytical considerations – that acid-hydrolysis can often result in an

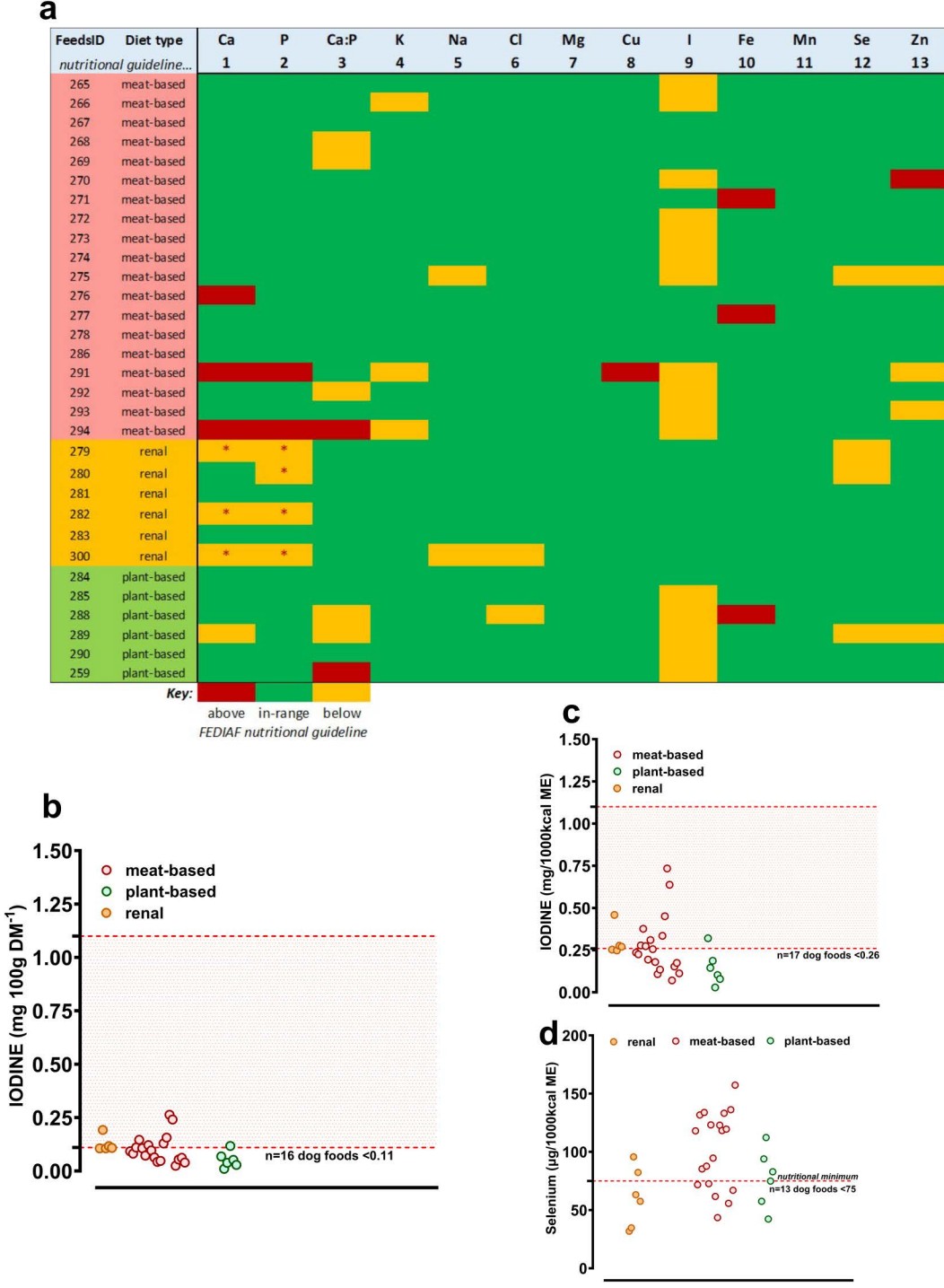

Fig 3. **Major and trace elemental content of dry feeds for dogs according to feed-type.** a) data expressed relative to the guideline nutritional minimum for each individual major and trace element ('Nutritional Guideline'; respective FEDIAF guideline #1 - #13). **Green** boxes represent values in range according to our analyses, **yellow** boxes are below nutritional minimum, **red** boxes are above nutritional (or Legal) maximum. **yellow** boxes with a * are allowable low values according to EU2020/354 intended use of feed for particular nutritional purpose (PARNUT). b,c,d, individual data points for measured iodine (b, mg/100g DM or c) mgs/1000kcal) and d) selenium between feed types with beef, lamb and chicken combined as 'meat-based'. Upper and lower dashed lines give relevant nutritional range for compliance.

**Table 2. Major and trace elemental content of dry feeds for dogs according to feed-type.**

| Major element (per 1000kcal) | FEDIAF (per 1000kcal) | Meat-based (n = 19) | Plant-based (n = 6) | Renal (n = 6) | Statistics *P-value |
|---|---|---|---|---|---|
| Calcium*1 | 1.25–6.25 | 4.68 ± 2.93[a] | 2.81 ± 1.13[ab] | 1.06 ± 0.24[b] | **0.009** |
| Phosphorus*2 | 1.0–4.0 | 3.16 ± 1.19[a] | 2.24 ± 0.53[a] | 0.77 ± 0.08[b] | **<0.001** |
| Ca:P ratio*3 | 1:1–2:1 | 1.41 ± 0.38 | 1.33 ± 0.63 | 1.40 ± 0.25 | 0.91 |
| Potassium*4 | 1.25 | 1.76 ± 0.39[a] | 2.54 ± 0.47[b] | 1.62 ± 0.10[a] | **0.001** |
| Sodium*5 | 0.25 | 1.05 ± 0.47 | 0.91 ± 0.25 | 0.56 ± 0.11 | 0.053 |
| Chloride*6 | 0.38 | 1.80 ± 1.00 | 1.60 ± 0.85 | 1.60 ± 0.34 | 0.85 |
| Magnesium*7 | 0.18 | 0.32 ± 0.05[a] | 0.50 ± 0.16[b] | 0.22 ± 0.02[c] | **<0.001** |
| **Trace element** ([mg or µg] per 1000kcal) | | | | | |
| Copper*8 | 1.80 | 4.55 ± 2.25 | 5.17 ± 1.47 | 3.62 ± 1.01 | 0.67 |
| Iodine*9 | 0.26 | 0.28 ± 0.17[a] | 0.14 ± 0.10[b] | 0.30 ± 0.09[a] | **0.02†** |
| Iron*10 | 9.00 | 84.3 ± 57.3[a] | 117 ± 99[a] | 34.1 ± 9.3[b] | **0.006†** |
| Manganese*11 | 1.44 | 12.4 ± 4.5 | 17.1 ± 8.2 | 13.4 ± 4.2 | 0.20 |
| Selenium (µg)*12 | 75.0 | 101 ± 33[a] | 77.3 ± 25.1[ab] | 60.8 ± 25.3[b] | **0.01** |
| Zinc*13 | 18.0 | 35.8 ± 11.6 | 34.3 ± 12.8 | 37.2 ± 10.7 | 0.91 |

Table 2. Major and trace elements for each dietary group, with FEDIAF nutritional guidelines given for reference. Values are mean ± SD and presented as g per 1000kcal (major elements) and mg or µg per 1000kcal (trace elements). *1-13, appropriate FEDIAF nutritional guideline. †, statistic by one-way ANOVA after $\log_{10}$ transformation of raw data.

under-estimate of its concentration [20]. Nevertheless, 9/10 of the EAAs for canines are reported: arginine, histidine, isoleucine, leucine, lysine, (methionine), phenylalanine, threonine and valine. One food had below the nutritional minimum guideline for 5 out of 9 EAAs tested. The longer-term effects of feeding such an EAA deficient diet to an animal with suspected CKD are not known. Current clinical guidelines for patients with CKD, without diabetes, advocate a diet low in protein but with supplemental keto-analogues of essential amino acids to support metabolic cycles dependent on supply of EAA [21]. In canines, if the same diet were fed for a long-period, then it is possible that other co-morbidities might be exacerbated; low intake of S-containing amino acids, for example, is associated with an increased risk of developing dilated cardiomyopathy, due to reduced taurine synthesis [22].

It was hypothesised that plant-based foods would contain inadequate branched-chain amino acids (BCAAs; leucine, isoleucine and valine), as most dietary BCAAs are derived from meat, fish and dairy products [23]. However, all meat- and plant-based foods met minimum nutritional requirements for BCAAs and average concentrations were, in fact, greater in plant-based foods compared to those comprised of predominantly beef or lamb. Again, one of the veterinary-renal foods had lower than recommended BCAA content. Unlike cats, the majority of dogs can synthesise taurine endogenously using sulphur-containing amino acids (e.g., methionine and cysteine, [24]). As such, taurine is not considered 'essential' for canines. For some large-breed dogs, such as Newfoundlands, taurine is essential, as a genetic mutation means they are unable to synthesis taurine endogenously and are therefore reliant on adequate dietary intake [25]. Taurine might, therefore, be considered a conditionally-essential amino acid for some breeds of dog and dietary choices, such as breed-specific foods, for such breeds should be made on a case-by-case basis. Other factors such as nutrient-nutrient interactions, amount of dietary fibre and fat-to-protein ratio in the gastrointestinal tract may also affect the bioavailability of other marginal AAs, limiting their uptake, particularly in those foods with only marginally-replete content [26]. Therefore, for all foods designated as 'complete,' an assigned nutritional minimum for all essential and conditionally-essential amino acids should be followed [27].

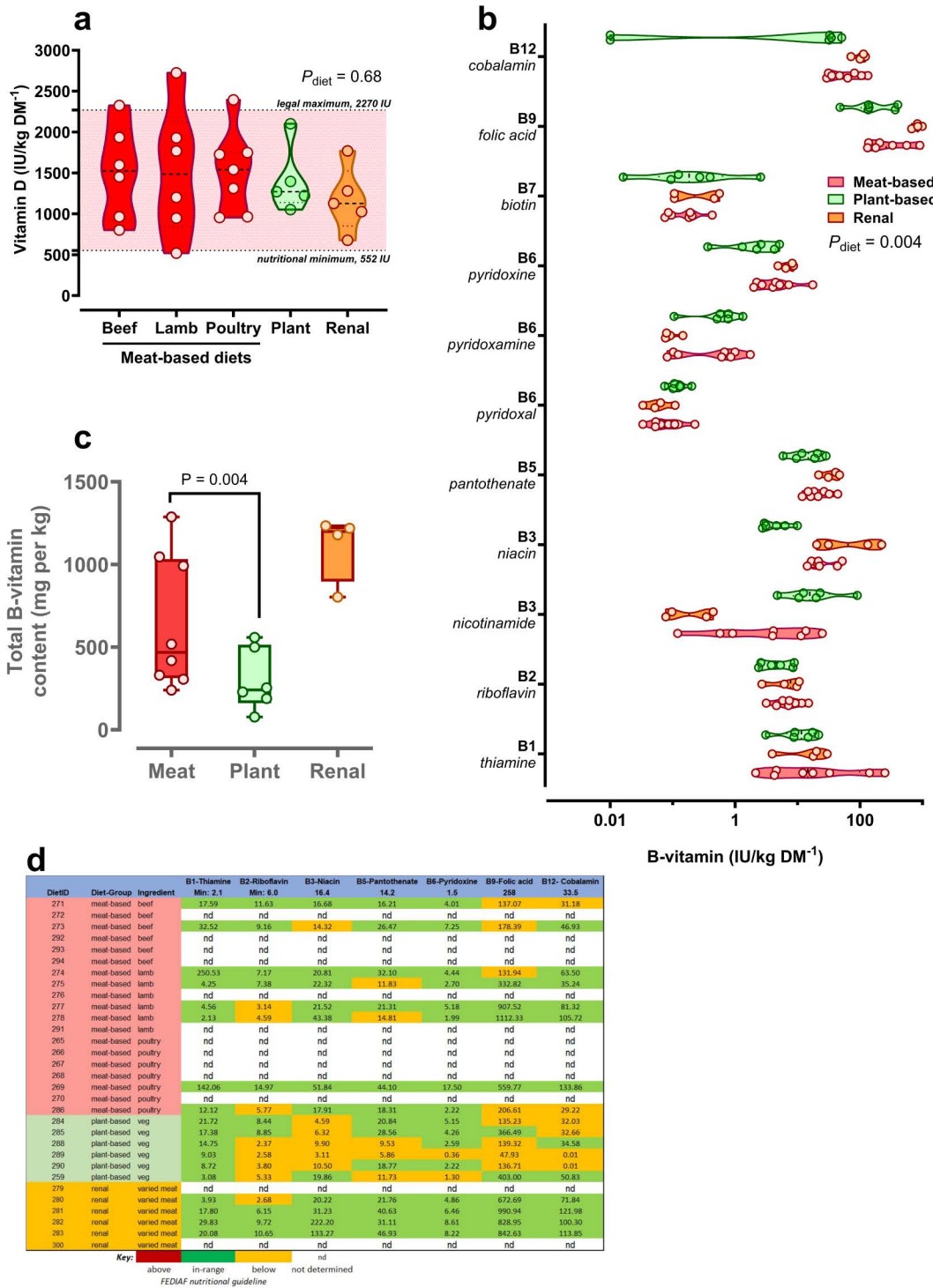

**Fig 4. Vitamin content (B1-B12, Vit D) of dry feeds for dogs according to feed-type.** (a) individual data points for measured vitamin D between feed-types, (b) individual data points for measured b-vitamins in food types, *P*-value by NP ANOVA (Kruskall-Wallis test), (c) individual data points for measured vitamin B9 (folic acid) in food types, note different scale on x-axis, (d), summated total B-vitamins between foods, *P*-value by NP ANOVA (Kruskall-Wallis test), (e), data expressed relative to the guideline nutritional minimum for each individual B-vitamin (top row). **Green** boxes represent values in range according to our analyses, **yellow** boxes are below nutritional minimum.

**Table 3. B-vitamin content of dry feeds for dogs according to feed-type.**

| B-vitamin | FEDIAF (per 1000kcal) | Meat-based (n = 8) | Plant-based (n = 6) | Renal (n = 4) | Statistics *P-value |
|---|---|---|---|---|---|
| Vitamin B1- Thiamine (mg) | 0.54 | 14.5 ± 22.6 | 3.58 ± 1.39 | 4.48 ± 2.67 | 0.89 |
| Vitamin B2- Riboflavin (mg) | 1.50 | 1.99 ± 0.97 | 1.30 ± 0.80 | 1.83 ± 0.91 | 0.33 |
| Vitamin B3- Niacin (mg) | 4.09 | 6.52 ± 3.43[ab] | 1.72 ± 0.81[b] | 25.4 ± 23.7[a] | **0.003** |
| Vitamin B5- Pantothenic acid (mg) | 3.55 | 5.79 ± 2.68 | 4.18 ± 2.27 | 35.1 ± 11.0 | 0.06 |
| Vitamin B6- Pyridoxine (mg) | 0.36 | 1.42 ± 1.27 | 0.73 ± 0.47 | 1.76 ± 0.43 | 0.07 |
| Vitamin B9- Folic acid (µg) | 64.5 | 111 ± 94[ab] | 41.3 ± 29.7[b] | 207 ± 32[a] | **0.02** |
| Vitamin B12- Cobalamin (µg) | 8.36 | 16.5 ± 9.6[ab] | 4.96 ± 4.54[b] | 25.5 ± 5.5[a] | **0.01** |
| **Total B-vits (mg, B1-B6)** | – | 30.2 ± 26.8[a] | 11.5 ± 4.5[b] | 42.3 ± 28.0[c] | **0.05** |
| **Total B-vits (µg, B9 + B12)** | – | 127 ± 101[a] | 46.2 ± 32.4[b] | 233 ± 37[c] | **0.01** |

Table 3. All data are expressed as units (mg or µg) per 1000kcal, based on 110kcal/kg maintenance energy requirement (MER). FEDIAF guideline content for individual vitamins given as appropriate, for reference. Data are mean ± SD, analysed by Kruskall-Wallis NP ANOVA.

**Fatty acids:** Few nutritional recommendations exist for fatty acids [9]. In this study, we were able to quantify n = 36 fatty acids (from C6:0 to C22:6n3) and report that where guidelines exist, (on a g per total lipid mass basis) all foods were replete in fatty acids, including those essential for canines (e.g., C18:2n6, linoleic acid), despite the fact that the majority of n-3 long-chain fatty acids such as eicosapentaenoic acid (EPA, 20:5) and docosahexanenoic acid (DHA, 22:6) are traditionally sourced from marine oils [28], which are not incorporated into plant-based foods. Alternative sources of omega 3- and 6 fatty acids for incorporation into plant-based foods, including chia and hemp seeds, flaxseed, walnuts, soya, seaweed, and microalgae, can be used satisfactorily [29]. Hence, with the varied nutritional sources as listed in decreasing order of incorporation on the labels in the foods tested here, then all requirements for essential and non-essential fatty acids were adequately met.

**Major and trace minerals:** Taken together, the current analysis of all major and trace elements in terms of compliance compared to relevant national guidelines for plant- and meat-based foods is similar to that reported previously [13]. That is, foods were broadly compliant (n = 342/391 [87%] of minerals were 'in-range'), but only 16% of foods satisfied all mineral guidelines. Veterinary-renal foods where certain minerals are exempted (e.g., Ca and P) due to being low for a 'particular nutritional purpose' ('PARNUTS', [30]) were not included in this analysis and were balanced in terms of the Ca:P ratio. Plant-based foods, in general, contained greater potassium and magnesium, consistent with mineral enrichment within plants, as well as sufficient in elemental iron – which can, for example, be limiting for female vegetarians [31]. In general, only sporadic deviations from recommended nutritional minimums or maximums were noted in some major minerals, e.g., calcium, phosphorous and potassium and some trace (often selenium or zinc) minerals. It is unlikely therefore that any clinical signs of malnutrition would develop as a result of these micronutrient imbalances, unless fed exclusively for long periods of time and over multiple batches of the same food.

Nevertheless, of particular note was that iodine was below the guideline nutritional minimum in over half (57%, n = 17 of 30) of all foods measured and remained when expressed on weight basis or corrected for energy density of foods. It was note-worthy that n = 5 of 6 plant-based foods measured had relatively low iodine, which is in-keeping with people following plant-based diets [18,32]. It would be relatively simple to supplement these foods with plant-based sources of high iodine, such as seaweed or sea-kelp [33]. Indeed, the only plant-based food with adequate iodine had both seaweed and dried algae as significant ingredients. A previous study by us reported similar results for feline diets, which also varied considerably between batches for iodine content [12]. Regardless, whilst iodine deficiency is common in vegetarians or vegans [34,35], few studies have reported any adverse effects of low iodine intake over the long-term in canines.

**Vitamin D:** Vitamin D insufficiency has been previously reported in dogs fed commercial meat-based [36] or home-made diets [37]. Ingestion of food, as opposed to sunlight, is the primary source of Vitamin D for canines [38]. In the current study, direct analysis of the foods for Vitamin D demonstrated none to be deficient. Vitamin D is a fat-soluble micronutrient with active endocrine properties [39]. The latter are particularly important during growth and development, given the important role Vitamin D has in calcium and phosphorus homeostasis; chronically low intake or deficiency of Vitamin D can cause bone de-mineralisation, through release of stored calcium and/or phosphate and may influence other non-skeletal related conditions [40]. Chronically elevated intake can lead to increased calcium and phosphorus absorption by the gut, with any subsequent hypercalcaemia being associated, in the longer-term, with chronic kidney disease [41]. Reporting values close to the upper nutritional guideline in pet food (one meat-based food was within 2% of the nutritional maximum) could become a problem if fed for a long period of time. Furthermore, considering the growth in plant-based food products for pets, it should also be noted that source of Vitamin D can influence bioavailability, which has not been measured here; active vitamin D2 (ergocalciferol) and D3 (1,25-dihydroxy-cholecalciferol) from meat-based sources are generally more bioavailable than vitamin-D2 from plant-based sources [42,43].

**B-vitamins:** B-vitamin deficiency can be caused by a number of gastrointestinal conditions in canines that, irrespective of dietary sufficiency, mean that B-vitamin uptake in the gut is reduced and less are bioavailable for cellular functions [44]. B-vitamin deficiency can have a range of effects on the body, including but not limited to, affecting the integrity of the dermal layer, acute disturbances of the central nervous system, lethargy, vomiting and diarrhoea [44,45]. Research into B-vitamin homeostasis in companion animals is lacking and complicated by variation according to breed [44]. B-vitamins are water-soluble and readily excreted in urine if taken in excess – which is common if predominantly consuming an animal-based diet. Hypervitaminosis, particularly of B-vitamins, is therefore rarely of clinical concern. In contrast, deficiency is commonly reported in vegetarians and vegans [19]. Regardless, all B-vitamins can easily be obtained from plant-based sources, although may need to be consumed in higher quantities due to poor bioavailability [43]. Consequently, humans often rely, at least partially, on supplementation from fortified foods, which may or may not be sufficient to meet dietary requirements [19]. A recent study found that Vitamin B12 status was similar between vegans (almost all of whom consumed supplements) and non-vegans (approx. 1/3 consumed supplements) [46]. For most foods analysed in this study – particularly plant-based, where the majority were lower in B1, B2, B3, B5, B9 and B12 – then further supplementation is recommended. Indeed, even for meat-based foods, B-vitamin supplementation using pre-mixes is common due to variability in B-vitamin content between animal products (e.g., muscle, organs) which can vary; cobalamin (vitamin B12) is low in muscle tissue for example [47]. In addition, possible losses of B-vitamins (e.g., A, D, E, C and B9 [folic acid]) can occur during the refinement process toward production of a dog kibble [48]. Increased temperature, for example, reduces active B-vitamin content in extruded foods [48].

**Overall compliance of pet foods:** Finally, all 31 complete dry dog foods were tested singularly and compared to European Pet Food Industry Federation (FEDIAF) guidelines [9]. When compared to FEDIAF guidelines, 17/31 foods tested met all amino acid, 5/31 met all mineral, 4/18 met B-vitamin and all tested met vitamin D guidelines. No food met FEDIAF guidelines for all nutrients. It is important to note that this analysis was conducted on complete, pre-digested food. There are many factors that will influence the uptake of nutrients within the body. Nevertheless, even nutrients that are replete in food may become deficient or have low bioavailability/bio-accessibility in the gastrointestinal tract due to nutrient-nutrient interactions.

**Conclusion and future directions:** Plant-based feeding of companion canines is becoming more common. It is important to reassure owners of such pets that feeding a 'complete' food, as designated on the label, does indeed mean feeding a food replete in all nutrients and micronutrients that are essential for dogs. Our study reports the first complete nutritional comparison of meat-based (including veterinary-renal) and plant-based foods for canines in the UK and suggests variable compliance: 55%, 100%, 16%, 100% and 24% of foods when compared to nutritional guidelines for amino acid, fatty acids, major and trace mineral, vitamin D and B-vitamins, respectively. No foods met ALL guidelines.

Nevertheless, one limitation of the current study is that it is only applicable to adult dog foods. Arguably, such micronutrient deficiencies would have greater impact if fed during growth & development or reproductive phases, when greater demand is placed on metabolic partitioning of nutrients. Further analyses of such foods, where available, is warranted. In addition, only n = 6 plant-based foods were tested. However at the time of study, this covered the majority of the UK plant-based pet food market. Since this time, other manufacturers have moved into the market and our conclusions would not automatically be representative of them. Plant-based foods were often low in iodine and many of the B-vitamins, which could be corrected by incorporation of mineral-rich ingredients and/or supplementation. Veterinary diets with lower protein content by design (e.g., for dogs with kidney disease) often had below-recommended levels of incorporation of essential amino acids, which could also be corrected through supplementation with keto-analogues of essential amino acids. Many of the remaining meat-based foods, were compliant but marginal, with respect to nutrient composition and may also benefit from supplementation. In this instance, further factors such as bioaccessibility, digestibility and nutrient-nutrient interactions may produce systemic micronutrient deficiencies, despite guidelines taking these factors into account when being established. Clearly, analysis of such multi-variate, gastrointestinal interactions is beyond the scope of the current study, but when novel foods are brought to market that might evidently have bioaccessibility effects (e.g., plant-based foods), then such studies are warranted.

## Supporting information

**S1 File.   S1 Table.** Standard Reference Material (SRM) transitions used for the AA standards, calibration standards and amino acid hydrolysate samples. **S2 Table.** uHPLC-MS/MS Instrumentation, Chromatographic and MS Parameters uHPLC Gradient for analysis of individual amino acids. **S3 Table.** Serine (g/100g DM) as determined in 30 of 31 samples of pet feed. **S4 Table.** Preparation of buffers for extraction of cytoplasmic (i.e., free) lipid from feeds using the 'sucrose cushion' method. **S5 Table.** LC-MS methods & conditions for determination of Vitamin D2 and D3. **S1 Fig.** Example trace for detection of Vitamin B12 in extracted pet food samples. **S6 Table.** LoQ values for B vitamin analysis.
(DOCX)

## Acknowledgments

The authors gratefully acknowledge the help of Dr Catherine Williams (NUVetNA Ltd) and Saul Velasquez for assistance with mineral analyses, Noriane Cochetal for assistance with amino acid data analysis, Drs Louise Williams and Jon Stubberfield for macronutrient analysis.

DSG conducts nutritional analysis for one of the tested companies' foods. D.S.G is not involved in the formulation of the food, nor has influence on the design or reporting of results. Since completing the analysis, R.A.B has purchased shares in the same company; testing was completed before this occurred.

## Author contributions

**Conceptualization:** Rebecca A. Brociek, David S. Gardner.

**Data curation:** Dongfang Li, Richard Broughton, David S. Gardner.

**Formal analysis:** Rebecca A. Brociek, David S. Gardner.

**Funding acquisition:** David S. Gardner.

**Investigation:** Rebecca A. Brociek, David S. Gardner.

**Methodology:** Rebecca A. Brociek, David S. Gardner.

**Project administration:** Rebecca A. Brociek, David S. Gardner.

**Software:** David S. Gardner.

**Supervision:** David S. Gardner.

**Validation:** Dongfang Li, Richard Broughton, David S. Gardner.

**Writing – original draft:** Rebecca A. Brociek, David S. Gardner.

**Writing – review & editing:** Rebecca A. Brociek, Dongfang Li, Richard Broughton, David S. Gardner.

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
