## [Decision Letter · Decision Letter 0]

12 May 2025

PONE-D-24-48404Nutritional analysis of commercially available, complete plant- and meat-based dry dog foods in the UKPLOS ONE

Dear Dr. Gardner,

Thank you for submitting your manuscript to PLOS ONE. After careful consideration, we feel that it has merit but does not fully meet PLOS ONE’s publication criteria as it currently stands. Therefore, we invite you to submit a revised version of the manuscript that addresses the points raised during the review process.

We look forward to receiving your revised manuscript.

Kind regards,

Ewa Tomaszewska, DVM Ph.D

Academic Editor

PLOS ONE

Journal Requirements:

2. Thank you for stating the following in the Competing Interests section: [DSG conducts nutritional analysis for one of the tested companies’ foods. D.S.G is not involved in the formulation of the food, nor has influence on the design or reporting of results. Since completing the analysis, R.A.B has purchased shares in the same company; testing was completed before this occurred].

4. Please include captions for your Supporting Information files at the end of your manuscript, and update any in-text citations to match accordingly. Please see our Supporting Information guidelines for more information: http://journals.plos.org/plosone/s/supporting-information .

Reviewers' comments:

Reviewer's Responses to Questions

**Comments to the Author**

1. Is the manuscript technically sound, and do the data support the conclusions?

Reviewer #1: Yes

Reviewer #2: Yes

2. Has the statistical analysis been performed appropriately and rigorously? 

Reviewer #1: Yes

Reviewer #2: Yes

3. Have the authors made all data underlying the findings in their manuscript fully available?

Reviewer #1: Yes

Reviewer #2: Yes

4. Is the manuscript presented in an intelligible fashion and written in standard English?

Reviewer #1: Yes

Reviewer #2: Yes

5. Review Comments to the Author

Reviewer #1: 1. Respectfully should read respectively line 7, para1 page 14

2. Page 22 para 1 line 15 Reword the following to remove word churlish “It is almost churlish to propose, therefore, that for all foods designated as ‘complete,’ an assigned nutritional minimum for all essential and conditionally-essential amino acids should at least be followed”

3. I think this reference should be dated 2021 not 2022. Also, these Guidelines have now been updated in 2024. FEDIAF. 2022. "Nutritional guidelines for complete and complementary pet food for cats and dogs." In. http://www.fediaf.org/self-regulation/nutrition.html.

Reviewer #2: Thank you for your interesting study, and helping to advance this field. I’ve recommended Minor revisions.

Major point

The conclusions that supplementation of plant-based diets should be considered—whereas this is usually not suggested for the other diets—is a little unbalanced, as (i) only a very small sample of dry, UK plant-based diets were assessed (so the claim may not be true of plant-based diets generally—anywhere, as it is currently worded, e.g. in the Abstract – this is partially acknowledged in the Conclusions), and (ii) similar suggestions concerning supplementation should be made for the other diets types also found to be deficient. This balance should be redressed where plant-based diets (only) are currently singled out as warranting supplementation.

It should be more prominently noted for context (where deficiencies or supplementation are described, including the Abstract), that varying nutritional deficiencies and imbalances are common across most diet types (including non-plant based, ie meat-based diets), and indeed that in this study no foods were found to be without flaws (e.g. “No food met FEDIAF guidelines for all nutrients.”)

Minor points

Introduction

Para 1 citation should be corrected to Vegan Society.

Discussion

Add citation to end of 1st sentence to https://journals.plos.org/plosone/article?id=10.1371/journal.pone.0291791

References

There are 2 references for the Vegan Society. They appear incomplete and inconsistently formatted.

Tables

To improve readability consider boldfacing all p<0.05 (as well as *) and adding a note to that effect.

Table 2

Fe

Mg

The Col. 1 values are very different from those for all diets – are these figs correct?

Table 2 notes:

Should [mg/µg] g per 1000kcal

be mg or µg per 1000kcal

Table 3

For total B-vits, make clearer you’re referring to vits B1-B6, B9 & B12 (as 1-6, 9+12 is not clear)

6. PLOS authors have the option to publish the peer review history of their article (what does this mean? ). If published, this will include your full peer review and any attached files.

**Do you want your identity to be public for this peer review?** For information about this choice, including consent withdrawal, please see our Privacy Policy .

Reviewer #1: **Yes: ** Mike Davies

Reviewer #2: No

---

## [Author Response · Author response to Decision Letter 1]

22 May 2025

Reviewer Comments:

Reviewer #1

1. Respectfully should read respectively line 7, para1 page 14

Changed. Page 7 line 7

2. Page 22 para 1 line 15 Reword the following to remove word churlish “It is almost churlish to propose, therefore, that for all foods designated as ‘complete,’ an assigned nutritional minimum for all essential and conditionally-essential amino acids should at least be followed”

Removed. Page 15 line 15

3. I think this reference should be dated 2021 not 2022. Also, these Guidelines have now been updated in 2024.

Updated reference to current (2024). Page 8 line 21

Reviewer #2

Thank you for your interesting study, and helping to advance this field. I’ve recommended Minor revisions.

Thank you to the reviewer for your kind words, in an often unkind arena. We have responded to your comments in turn below.

Comment: supplementation of plant-based foods.

We note your points and have modified the text accordingly, in places (see marked-up red). I do not think it would come as any surprise that vegetarian/vegan pet foods, in line with plant-based human dietary patterns, would be low in some B-vits and iodine. These were consistent in all our, albeit small, sample of plant-based foods. Thus, we suggest that certain supplementation could be considered, without side effects. For the other diets, then isolated – but not consistent – deficiencies were noted (e.g. some essential amino acids in renal foods) and we have noted the overall lack of compliance throughout the manuscript, including the discussion. Indeed, we do agree with your point and have modified the text accordingly.

To do: It should be more prominently noted for context (where deficiencies or supplementation are described, including the Abstract), that varying nutritional deficiencies and imbalances are common across most diet types (including non-plant based, ie meat-based diets), and indeed that in this study no foods were found to be without flaws (e.g. “No food met FEDIAF guidelines for all nutrients.”

Para 1 citation should be corrected to Vegan Society

Changed Page 3, line 5

Add citation to end of 1st sentence

added

References

It is how endnote formats non-standard references. Will try to alter, but can do after the fact by hand.

Tables

Readability hopefully improved.

Table 2

Yes, values for iron and magnesium as mgs per 1000kcal according to FEDIAF 2024 are indeed 9.00 and 0.18 (mgs per 1000 kcal ME), respectively. Page 16 of 98 in FEDIAF 2024.

Table 2 notes:

Yes, have altered to improve readability

Table 3

Added ‘B’ to make clearer

---

## [Decision Letter · Decision Letter 1]

5 Jun 2025

PONE-D-24-48404R1Nutritional analysis of commercially available, complete plant- and meat-based dry dog foods in the UKPLOS ONE

Dear Dr. Gardner,

Thank you for submitting your manuscript to PLOS ONE. After careful consideration, we feel that it has merit but does not fully meet PLOS ONE’s publication criteria as it currently stands. Therefore, we invite you to submit a revised version of the manuscript that addresses the points raised during the review process.

We look forward to receiving your revised manuscript.

Kind regards,

Ewa Tomaszewska, DVM Ph.D

Academic Editor

PLOS ONE

Journal Requirements:

Reviewers' comments:

Reviewer's Responses to Questions

**Comments to the Author**

1. If the authors have adequately addressed your comments raised in a previous round of review and you feel that this manuscript is now acceptable for publication, you may indicate that here to bypass the “Comments to the Author” section, enter your conflict of interest statement in the “Confidential to Editor” section, and submit your "Accept" recommendation.

Reviewer #2: (No Response)

2. Is the manuscript technically sound, and do the data support the conclusions?

Reviewer #2: Yes

3. Has the statistical analysis been performed appropriately and rigorously? 

Reviewer #2: Yes

4. Have the authors made all data underlying the findings in their manuscript fully available?

Reviewer #2: Yes

5. Is the manuscript presented in an intelligible fashion and written in standard English?

Reviewer #2: Yes

6. Review Comments to the Author

Reviewer #2: Thank you for addressing all my previous suggestions. My only remaining suggestion is to review boldfacing applied in tables where p < 0.05. A very few instances where p < 0.05 seem to have been missed.

7. PLOS authors have the option to publish the peer review history of their article (what does this mean? ). If published, this will include your full peer review and any attached files.

**Do you want your identity to be public for this peer review?** For information about this choice, including consent withdrawal, please see our Privacy Policy .

Reviewer #2: No

---

## [Author Response · Author response to Decision Letter 2]

25 Jun 2025

i have made bold the three instances in tables, as suggested by reviewer #002. No other changes requested.

---

## [Editor Report · Decision Letter 2]

2 Jul 2025

Nutritional analysis of commercially available, complete plant- and meat-based dry dog foods in the UK

PONE-D-24-48404R2

Dear Dr. David S. Gardner,

We’re pleased to inform you that your manuscript has been judged scientifically suitable for publication and will be formally accepted for publication once it meets all outstanding technical requirements.

Kind regards,

Ewa Tomaszewska, DVM Ph.D

Academic Editor

PLOS ONE
---

## [Editor Report · Acceptance letter]

PONE-D-24-48404R2

PLOS ONE

Dear Dr. Gardner,

I'm pleased to inform you that your manuscript has been deemed suitable for publication in PLOS ONE. Congratulations! Your manuscript is now being handed over to our production team.

Kind regards,

on behalf of

Professor Ewa Tomaszewska

Academic Editor

PLOS ONE